# Metabolomics of Cerebrospinal Fluid Amino and Fatty Acids in Early Stages of Multiple Sclerosis

**DOI:** 10.3390/ijms242216271

**Published:** 2023-11-13

**Authors:** Michal Židó, David Kačer, Karel Valeš, Denisa Zimová, Ivana Štětkářová

**Affiliations:** 1Department of Neurology, Third Faculty of Medicine, Charles University, 100 00 Prague, Czech Republic; zido.michal@gmail.com; 2Department of Neurology, Faculty Hospital Královské Vinohrady, 100 34 Prague, Czech Republic; denisa.zimova@fnkv.cz; 3National Institute of Mental Health, 250 67 Klecany, Czech Republic; david.kacer@nudz.cz (D.K.); karel.vales@nudz.cz (K.V.); 4Department of Psychiatry and Medical Psychology, Third Faculty of Medicine, Charles University, 100 00 Prague, Czech Republic

**Keywords:** multiple sclerosis, MS, cerebrospinal fluid, CSF, metabolomics, arginine, histidine

## Abstract

Multiple sclerosis (MS) is a demyelinating and neurodegenerative autoimmune disease of the central nervous system (CNS) damaging myelin and axons. Diagnosis is based on the combination of clinical findings, magnetic resonance imaging (MRI) and analysis of cerebrospinal fluid (CSF). Metabolomics is a systematic study that allows us to track amounts of different metabolites in a chosen medium. The aim of this study was to establish metabolomic differences between the cerebrospinal fluid of patients in the early stages of multiple sclerosis and healthy controls, which could potentially serve as markers for predicting disease activity. We collected CSF from 40 patients after the first attack of clinical symptoms who fulfilled revised McDonald criteria of MS, and the CSF of 33 controls. Analyses of CSF samples were performed by using the high-performance liquid chromatography system coupled with a mass spectrometer with a high-resolution detector. Significant changes in concentrations of arginine, histidine, spermidine, glutamate, choline, tyrosine, serine, oleic acid, stearic acid and linoleic acid were observed. More prominently, Expanded Disability Status Scale values significantly correlated with lower concentrations of histidine. We conclude that these metabolites could potentially play a role as a biomarker of disease activity and predict presumable inflammatory changes.

## 1. Introduction

Autoimmune diseases and their increasing incidence in the world’s population present severe problems for modern medicine, and multiple sclerosis is one of them. Multiple sclerosis (MS) is a serious inflammatory and neurodegenerative autoimmune disease targeting the central nervous system (CNS). MS is an incurable disease, but thanks to modern therapeutic procedures we are able to stabilize it for a certain period of time and increase the duration of the patient’s life with minimal disability. MS diagnosis is based on three important components: objective clinical findings, brain and spinal cord MRI and biochemical cerebrospinal fluid (CSF) analyses. The possibilities of treatment are vast. There are many different drugs available and newly in development, but their effectiveness and side effects can significantly differ. For one patient, a certain drug can stabilize the disease for years, but for another it may be ineffective or even dangerous because of often life-threatening side effects [1]. Because of this, an individual approach to the patient is preferred in the treatment of MS [2]. For this reason, it is also crucial to identify biomarkers for individual phases of the disease.

Cerebrospinal fluid (CSF) is a clear fluid, partially produced by the choroid plexus and ependymal cells of cerebral ventricles and partially by ultrafiltration of plasma within capillaries of the choroid plexus. The CSF composition resembles the blood’s plasma but is different in the concentrations of its components. CSF can reflect the state of the brain and blood–brain barrier, especially in cases of inflammatory changes. CSF plays an important role in the diagnosis of MS and additionally represents a medium to identify new potential MS biomarkers of MS [3]. The main advantage of CSF in laboratory use is that it does not need consistent adjustment, especially the precipitation of proteins needed when working with the blood’s plasma. 

Metabolomics is a systematic study that uses the approach of analytical chemistry to endogenously profile small metabolite molecules present in the investigated medium (in our case in the CSF). Metabolomics allows us to track the amount of different metabolites, which are parts of different cells and metabolic processes [4]. The comparison of metabolomic profiles of MS patients with healthy controls presents an important strategy in understanding the pathology of various diseases and responses to therapy. These metabolites can become new potential biomarkers of diagnosis, prognosis, or treatment efficacy for different diseases. Metabolomic studies use different analytical approaches. The important factor is if it is a non-targeted or targeted approach [5]. The targeted approach is focused on a certain metabolite or group of metabolites. The non-targeted approach observes the whole available spectrum of the metabolome in a medium. The most applied methods in metabolomic studies are nuclear magnetic resonance and mass spectrometry with liquid or gas chromatography. It is necessary to perform a thorough subsequent validation of findings by comparison with known analytical standards to confirm the identity of metabolites. 

The effort to treat MS as early as possible with the best suitable treatment is met with a shortage of useful prognostic biomarkers. Therefore, the need for new biomarkers of MS is still great, especially those that can help us predict the course of the disease and treatment efficacy. The metabolomics represent one of the strong potential sources of new biomarkers for multiple sclerosis. Several authors have in recent years published studies considering metabolomics in multiple sclerosis patients. Their results show significant differences in the composition and concentration of metabolites in CSF of MS patients in comparison with a control group. However, studies do not differentiate between the disease stages. The most common changes were observed in cases of amino acids and fatty acids. Some authors found decreased levels of tyrosine, leucine/isoleucine, arginine and phenylalanine in the CSF [6,7,8,9,10,11,12,13,14] and sera of MS patients [15]. Other authors described significant changes in levels of fatty acids in the CSF of MS patients [16,17]. The most interesting finding was in the case of glutamate, the levels of which increased in the CSF of MS patients with active lesions [11,18,19]. Glutamate changes were also observed in sera [20,21]. In cases of myoinositol and choline, the published results were inconclusive. Some authors found that their levels increased in MS patients [6,22,23]; however, others described conflicting results [24]. In one interesting study, the authors found that fingolimod affects the metabolism of some amino acids and presented their observation as a potential biomarker of fingolimod’s efficacy [25]. In one of our previous articles, we published pilot data consisting of significant changes in three metabolites (arginine, histidine and palmitic acid) in a small group of MS patients [26].

The main goal of our study was to establish metabolomic differences between the cerebrospinal fluid of patients in the early stages of multiple sclerosis and healthy controls and then try to explain these changes from a pathophysiological point of view. We also correlated significantly changed metabolites with EDSS values at the time of CSF collection, and 1 and 2 years after. These metabolite values were also correlated with the duration of the disease symptoms before CSF collection. 

## 2. Results

In total, 73 subjects (57 females, 16 males) participated in this study with a median age of 34 years (range 18–54 years). All subjects had no medical history of other neurological or autoimmune diseases that could potentially change the CSF composition. In addition, none of the subjects had a history of psychopharmacological drug use, which could also change the CSF composition. From the MS patient group, six had a history of hormonal anticonception use (ID 5, 11, 13, 14, 21, 34), another five patients took levothyroxinum daily (ID 6, 10, 12, 27, 38) and two other patients took antihypertensive medication daily (ID 12, 36).

Forty patients (31 females, 9 males) after the first symptoms with an MRI finding and CSF finding corresponding to demyelinating disease, type MS, fulfilling the revised McDonald criteria from 2017 [27], yet without any specific DMD treatment, were included into the “MS group”. All patients from the MS group had normal levels of protein in the CSF with an average value of 0.32 g/L (range 0.11–0.49 g/L), normal levels of glucose in the CSF with an average value of 3.3 mmol/L (range 2.94–5.5 mmol/L) and an acceptable mononuclear cell count with an average value of 12/μL (range 1–43/μL). All patients from the MS group additionally had positive oligoclonal bands (OCB) in the CSF—an IgG OCB average value of 7 (range 0–18), IgA OCB average value of 6 (range 0–16), IgM OCB average value of 6 (range 0–22), light chains kappa OCB average value of 3 (range 0–16) and light chains lambda OCB average value of 2 (range 0–12). Additionally, all of the MS group members had negative antibodies against MOG and AQP-4 in sera and CSF, respectively. Details of the basic cytological and biochemical analyses of the CSF are presented in Appendix A. CSF was on average collected 53 days after the onset of the clinical symptoms, except for in four subjects (ID 10, 19, 21, 25), whose CSF was collected in a range of 2 to 5 years from the onset of clinical symptoms, and in another two subjects (ID 7, 32) for which this information was not available. Thirty-one patients were observed in the Center for Demyelinating Diseases and the Neurological Department of the third Faculty of Medicine, Charles University and Teaching Hospital Kralovske Vinohrady, after 1 year, and 21 of them also after 2 years, where they were regularly examined with EDSS values by experienced neurologists. EDSS values (average value of 1.8, range 0.5–3.5) were measured in all 40 subjects at the time of CFS sample collection. EDSS values were evaluated in 31 subjects after 1 year, with an average value of 2.0 (range 0.5–4.0). EDSS values were measured in 21 subjects after 2 years, with an average value of 1.9 (range 0.5–3.0). Details of clinical information about the MS group patients are summarized in Appendix A. 

The control group of 33 subjects (26 females, 7 males) comprised patients with non-specific subjective difficulties, for example sensory disturbances, headaches or dizziness, whose clinical and paraclinical findings were normal, not corresponding to any specific neurological disorder (“symptomatic controls” by Teunissen et al., 2013 [28]).

### 2.1. Metabolomic Analyses

Through untargeted metabolomic analyses we found around 60 different metabolites in CSF. The most significant changes were observed in amino and fatty acids in the CSF of patients after the first clinical symptom of MS in comparison to the control group. The most statistically significant changes were found in 12 metabolites—arginine, histidine, spermidine, glutamate, choline, tyrosine, serine, methionine, homovanillic acid, linoleic acid, oleic acid and stearic acid. In 10 metabolites we were able to count their exact concentrations using analytical standards. Statistical details about the concentrations of these 10 metabolites are summarized in Table 1, and the statistical details of 2 other significant and 18 non-significant metabolites are present in Table 2. Complete descriptive statistical details are presented in the Appendix A.

The most pronounced observation in the MS group was a statistically significant decrease of various metabolomics, except for one with significant increase (spermidine). Statistically significant changes were observed in arginine (*p*-value: 0.003), the value of which decreased in the MS group (average concentration of 3.97 μg/mL) in comparison to the control group (average concentration of 5.24 μg/mL) (Figure 1A). Histidine (*p*-value: 0.005) decreased in the MS group (average concentration of 7.02 μg/mL) in comparison to the control group (average concentration of 8.16 μg/mL) (Figure 1B). Glutamate (*p*-value: 0.015) also decreased in the MS group (average concentration of 1.89 μg/mL) in comparison to the control group (average concentration of 2.12 μg/mL) (Figure 1D), as did choline (*p*-value: 0.023), which decreased in the MS group (average concentration of 1.04 μg/mL) in comparison to the control group (average concentration of 1.23 μg/mL) (Figure 2A). Other significant changes were observed in tyrosine (*p*-value: 0.031), which decreased in the MS group (average concentration of 1.10 μg/mL) in comparison to the control group (average concentration of 1.25 μg/mL) (Figure 2B), and serine (*p*-value: 0.047), which decreased in the MS group (average concentration of 2.27 μg/mL) in comparison to the control group (average concentration of 2.59 μg/mL) (Figure 2C). Only spermidine (*p*-value: 0.012) showed a significant increase in the MS group (average concentration 0.04 μg/mL) in comparison to the control group (average concentration of 0.03 μg/mL) (Figure 1C). Significant changes were also observed in some fatty acids, particularly linoleic acid (*p*-value: 0.001), which decreased in the MS group (average concentration of 3.13 μg/mL) in comparison to the control group (average concentration of 3.39 μg/mL) (Figure 2D). On the other hand, oleic acid (*p*-value: 0.015) increased in the MS group (average concentration of 8.79 ng/mL) in comparison to the control group (average concentration of 8.39 μg/mL) (Figure 3A). Stearic acid (*p*-value: 0.029) decreased in the MS group (average concentration of 2.77 ng/mL) in comparison to the control group (average concentration of 3.15 μg/mL) (Figure 3B).

In other cases, we could not establish precise concentrations but, nevertheless, the next two metabolites were still significantly changed in patients after the first clinical symptoms of MS in comparison to the control group. For methionine (*p*-value: <0.001), we observed increased levels in the MS group (average of 8504.86) in comparison to the control group (average of 6256.72) (Figure 4A). For homovanillic acid (*p*-value: 0.023), we observed decreased levels in the MS group (average of 25,611.18) in comparison to the control group (average of 29,774.24) (Figure 4B). Differences in levels of other amino acids like threonine (*p*-value: 0.053, Figure 4C), leucine/isoleucine (*p*-value: 0.223, Figure 4D), alanine (*p*-value: 0.262) or aspartate (*p*-value: 0.541) were not statistically significant in the MS group in comparison to the control group. In comparison to results from our pilot study [21], we did not find any significant changes in palmitic acid (*p*-value: 0.711) here.

### 2.2. Correlation of Metabolomics Results with EDSS Values

Results of metabolomics analyses were correlated with EDSS values after the first clinical symptoms of multiple sclerosis at the time of CSF collection (40 patients), 1 year after (31 patients) and 2 years after (21 patients).

In histidine, we did not find any significant correlation between its concentrations and EDSS values at the time of CSF collection (r: 0.05; *p*-value: 0.74). One year after the CSF collection, we observed a statistically significant medium–strong negative correlation with histidine (r: −0.42; *p*-value: 0.03) (Figure 5). In EDSS values 2 years after, we also observed a statistically significant medium–strong negative correlation with histidine (r: −0.43; *p*-value: 0.048).

In arginine, spermidine, glutamate, choline, tyrosine, serine, linoleic acid, oleic acid and stearic acid we did not find any statistically significant correlation with EDSS values at the time of CSF collection, nor with EDSS values after 1 year and after 2 years. The statistical details of individual correlations are described in Table 3.

### 2.3. Correlation of Metabolomic Results with Onset of Clinical Symptoms to CSF Collection

We correlated metabolomic analyses with the duration of clinical symptoms at the time of MS group patient CSF collection. This information was available for 38 patients. 

In arginine (r: 0.07; *p*-value: 0.70), histidine (r: 0.03; *p*-value: 0.84), spermidine (r: 0.19; *p*-value: 0.27), glutamate (r: 0.01; *p*-value: 0.94), choline (r: 0.20; *p*-value: 0.25), tyrosine (r: −0.20; *p*-value: 0.23), serine (r: 0.06; *p*-value: 0.75), linoleic acid (r: 0.11; *p*-value: 0.54), stearic acid (r: 0.13; *p*-value: 0.47) and oleic acid (r: −0.14; *p*-value: 0.42) we did not observe any statistically significant correlation between their concentrations and the duration of clinical symptoms at the time of CSF collection.

## 3. Discussion

Our study primarily aimed to identify variances in metabolites within the cerebrospinal fluid of individuals in the initial phases of multiple sclerosis as compared to those who are healthy. Additionally, we aimed to elucidate the underlying reasons for these alterations from a pathophysiological perspective. Furthermore, we conducted correlations between the notably altered metabolites and the EDSS values during the point of CSF sampling, as well as at 1 and 2 years following the collection.

In this study, we have proven statistically significant changes in the metabolomics of cerebrospinal fluid in MS patients after the first clinical symptoms in comparison to healthy controls. The most significant changes were the observed decreases in arginine, histidine, glutamate, choline, tyrosine, serine, methionine, homovanillic acid, linoleic acid and stearic acid. Only spermidine and oleic acid showed a statistically significant increase in the MS group. These metabolites have great potential and could serve as new potential biomarkers for the early stages of MS. However, these observations have yet to be verified and confirmed in further clinical studies.

Several authors have already published their observations concerning the metabolomics of CSF in patients with MS [6,7,8,9,13,14,16,17,22,23,24,25], but their results frequently came from samples of non-homogenous patient groups. Those groups comprised MS patients in different stages of the disease, more often in the later stages, with different DMD treatments and medical and pharmacological histories, which could potentially interfere with the CSF composition within those patients. In our study we created a strictly homogenous MS patient group and control group. In the MS group we focused on patients in the early stages of MS after the first clinical symptoms of MS, with no history of psychopharmacological drug use, yet without specific DMD or high-dose corticosteroid treatment. In some publications, patients with different neurological conditions were part of the control group, in some cases even with inflammatory neurological conditions like encephalitis, which could potentially affect the results. In our study, the control group was based on the definition from a 2013 publication [28] and comprised patients with subjective non-specific difficulties like sensory disturbances, headaches or dizziness, with negative clinical and paraclinical findings, not corresponding to any certain neurological disease. We consider our effort to create a strictly homogenous MS patient group in the early stages of the disease and controls without any interfering factors a strong factor of our study. One limitation of this study could be the relatively small patient sample size.

Our results show significant alterations in amino and fatty acid concentrations in the CSF of early-stage MS patients compared to healthy controls. The first finding is changes in L-arginine. L-arginine is an amino acid and a precursor of nitric oxide (NO), an important neurotransmitter, which is also involved in various metabolic pathways [29]. In places of inflammation, i.e., active/acute lesions in MS, a special version of enzyme called an inducible NO synthase (iNOS) is activated. It produces massive concentrations of NO independently from calcium concentrations [30]. In MS, more precisely in the regions of active lesions, higher concentrations of NO were found [31]. Higher concentrations of NO were also observed in CSF and sera [32]. Based on this, we can assume that decreased levels of arginine can reflect increased precursor depletion within ongoing inflammation, where NO is being created in massive concentrations with the help of iNOS. Another factor contributing to L-arginine depletion may be increased activity of the human arginase 1, cytosolic protein, in MS [33]. In this study, we did not focus on NO, but through the untargeted approach we found significantly decreased levels of arginine in patients after the first clinical symptoms of MS in comparison to the controls. This finding is in agreement with findings of several other authors [6,7,8,9]. We can hypothesize that arginine can potentially represent a biomarker of disease activity, more precisely of inflammatory activity, especially in the early stages of the disease. In this moment we cannot confirm its specificity for MS; therefore, the next steps would be to compare this finding with other types of CNS inflammations such as autoimmune or infectious encephalitis.

Also, we observe findings in some additional amino acids. The amino acid histidine is a precursor of histamine. Histamine is involved in inflammatory processes and in the pathogenesis of MS, but the nature of this role is still unknown [34,35]. It is assumed, based on animal models, that elevated levels of histamine are connected to the increased synthesis of pro-inflammatory cytokines, TNF and interferon gamma [36]. In theory, histamine and histidine are involved in the fatigue of MS patients [37]. One study found decreased serum levels of histidine in fatigued female MS patients in comparison with non-fatigued female MS patients [38]. Several studies found increased levels of histamine in the CSF of MS patients [39,40]. Decreased histidine levels may be explained by its higher need as a precursor for the synthesis of histamine. In this study, we observed a decreased level of histidine in MS patients. This finding is in agreement with other authors [5,6,7,8,9,10,11]. In addition, we observed a medium–strong negative correlation between concentrations of histidine and EDSS values 1 year and 2 years after the CSF analysis. We may speculate about some prognostic potential of histidine in the early stages of MS, more precisely about its ability to predict EDSS in the following 1 or 2 years of the disease. Lower concentrations of histidine can indicate a worse course with more pronounced deterioration and stronger fatigue symptoms.

Natural polyamine spermidine is essential for the proliferation, differentiation and survival of cells [41]. In connection with MS, spermidine has thus far only been described in experimental animal models [42]. In this study, we found significantly increased levels of spermidine in MS patients in comparison to the controls. Spermidine inhibits the synthesis of pro-inflammation cytokines and promotes the autophagy of damaged cells [43]. Based on this, increased levels of spermidine could be explained as a defensive reaction of the organism against autoimmune inflammation. However, the problem that the relationship between spermidine and autoimmune inflammation has not as yet been fully clarified remains.

Glutamate excitotoxicity is one of the pathological factors in many neurological diseases, as well as in MS [44]. We can speculate, based on this information, that glutamate levels could be a marker of axonal degeneration in MS. Several studies have shown increased glutamate levels in the CSF of MS patients [17,19]. In our study we found opposing results with significantly decreased glutamate in the CSF of MS patients. A potential explanation for this difference could be that our targeted MS patient group was in the early stages of the disease, when the process of demyelination is dominant over axonal degeneration. Some authors focused on the later stages of MS, when the process of axonal degeneration may be more prominent [17,19]. During axonal neurodegeneration, the destruction of axons and neurons occurs, potentially releasing high amounts of glutamate into the extracellular space and CSF [45]. Based on this, we can hypothesize that increased levels of glutamate are more typical of the later stages of MS, or of cases of MS where the process of neurodegeneration is stronger, and its levels may hypothetically reflect the process of neurodegeneration.

We observed a decreased level of choline in MS patients, which is in agreement with other authors [24]. Some authors published different results [6,22,23]. We may speculate that choline levels can reflect the capacity of remyelination in each individual patient, which could explain these differences. The effect of CDP-choline as a means of support for the process of remyelination has only been tested in animal models, so far with promising results [46]. 

The amino acids serine and tyrosine, having a neuroprotective function [47,48], have been observed by other authors to be decreased in the CSF of MS patients [9]. Serine participates in the regulation of certain anti-inflammation cytokines in MS and, with that, helps with the protection of myelin and supports remyelination [49]. Meanwhile, tyrosine plays an important role in the development, proliferation and regeneration of neurons [48]. Decreased levels of serine and tyrosine may reflect greater susceptibility to myelin damage, to demyelination in general and to worse regenerative capacities of the CNS. In this study, we found decreased levels of serine and tyrosine in our MS patients, which is in agreement with others.

The myelin sheet is an essential structure for the correct and fast neuronal transmission of nerve impulses [50]. Myelin mainly comprises lipids and proteins and their basic fatty acid components like oleic acid, linoleic acid and stearic acid. In MS, through the process of demyelination, myelin is deconstructed into its basic components by pathophysiological processes, which can hypothetically lead to increased levels of fatty acids in extracellular space and CSF. During the process of remyelination, we can expect decreased levels of fatty acids in extracellular space and CSF, as they are being used by new oligodendrocytes in their attempt to recreate and repair the proper myelin sheet [51]. Based on these facts, we may speculate that levels of fatty acids reflect demyelination or remyelination in MS patients. Increased levels of fatty acids point out demyelination, while decreased levels of fatty acids reflect remyelination [50]. Some authors have reported increased oleic, linoleic and stearic acids in the CSF of MS patients [7,17,50]; however, others have published opposing results with a decrease of fatty acids [16]. Our findings are in agreement with these authors, but in disagreement with the aforementioned. These different findings could be explained by the heterogeneity of patients with MS in the early or later stages of the disease with various representations of demyelination and remyelination. In our study, we focused on naïve MS patients without specific treatment, when the process of remyelination may still be strong and result in a decreased level of fatty acids in CSF. 

Cerebrospinal fluid presents as one of the most important biological materials that can help us understand the pathology of MS. CSF can be used to detect various analytes or cell populations, which is already used for diagnostic purposes. CSF can additionally be considered as a “golden standard matrix” in the diagnosis of MS [52]. Its main advantage is that it reflects the actual state of the CNS during inflammatory processes and, thanks to the small concentrations of proteins, does not need a special preparation before laboratory analyses. On the other hand, its main disadvantage is that CSF collection is an invasive procedure and is only performed on rare occasions.

Some authors have investigated the metabolomics of sera in MS patients [13,15,53] and found decreased levels of amino acids, more precisely in phenylalanine, tyrosine and tryptophan [13,15]. Other authors have published significant decreases of asparagine and carnitine [53]. In our study, we solely focused on CSF and, therefore, cannot relevantly evaluate and compare these results from sera.

## 4. Methods

### 4.1. Recruitment of Patients

The patients were recruited from the database of the Center for Demyelinating Diseases and the Neurological Department of the third Faculty of Medicine, Charles University and Teaching Hospital Kralovske Vinohrady, Prague, Czech Republic. In total, we recruited 73 patients, aged between 18 and 55 years, and divided them into two groups (MS patient group and control group). The basic inclusion criteria for both groups were no medical history of other neurological autoimmune diseases, no history of psychopharmacological drug use and aged between 18 and 55 years.

The first group comprised 40 patients (31 female, 9 male) after the first clinical symptoms of multiple sclerosis fulfilling the revised McDonald criteria from 2017 [29], with an MRI finding and CSF finding appropriate to demyelinating disease, type MS, yet without specific “disease-modifying-drug (DMD)” treatment. In addition, they all had negative findings of MOG and AQP-4 antibodies in their CSF and sera, the positivity of which could suggest the presence of other demyelinating diseases.

The second group comprised 33 control subjects (26 female, 7 male) based on the definition of so-called “symptomatic controls” published in 2013 [52]. Patients from this group suffered from different subjective difficulties, for example, non-specific dizziness, headaches, sensory disturbances like paraesthesia or dysesthesia, and others with normal clinical and paraclinical findings that do not correspond to any specific neurological diseases. 

All patients participating in this study signed informed consent and received a full explanation of the study and its character. All participants underwent a complete clinical neurological examination, cerebral and spinal cord MRI and a basic biochemical and cytological examination of their CSF and sera. 

### 4.2. Sample Collection and Preparation

The CSF sample was collected by performing a lumbar puncture under sterile conditions in the intervertebral space between the 5th and 4th lumbar vertebra into polypropylene tubes. Basic biochemical and cytological analyses were performed after the collection. Samples with an erythrocyte count higher than 600 in μL, pleocytosis above 50 in 1 μL or elevated protein levels over 0.5 g/L were excluded from the study. In the next step, samples were centrifuged at 3500/min for 4 min and frozen at a −80 °C temperature. Samples were stored and transported for laboratory analyses within the National Institute for Mental Health, Klecany, Czech Republic under these conditions.

Preparation of the analytical samples for the untargeted metabolomic analysis and targeted analysis of selected small molecular metabolites (50–300 Da) proceeded as follows: 100 μL of CSF was transferred into a precooled Eppendorf tube (1.5 mL), followed by the immediate addition of 400 μL of an ice-cold ACN:MeOH mixture (1:1, *v*/*v*) with internal standard (Deuterated Amino Acid Standard Mixture, Sigma Aldrich). Samples were vortexed for 30 s and consequently incubated for 1 h at −20 °C. Incubation was followed by 10 min of centrifugation at 13,000 rpm at 4 °C, and supernatants were transferred to Eppendorf tubes (0.5 mL) and evaporated to dryness using a SpeedVac concentrator. The dry extract was reconstituted in 100 μL of H2O:MeOH (1:1, *v*/*v*) and sonicated in an ice-cold bath for 20 min. The supernatant was transferred into a vial and directly analyzed.

Samples for analysis of fatty acids were thawed on ice, vortexed for 15 s and added with 100 μL of CSF to 400 μL of propan-2-ol (=IPOH) with additional internal standard prior to placement in an ultrasound bath for 10 min. Extraction was performed for 30 min at 4 °C, followed by centrifugation at 4000 rpm for 10 min at 4 °C. Supernatant (450 μL) was collected and dried under a nitrogen stream. Samples were reconstituted with 100 μL of ethanol (=EtOH), vortexed for 10 s, and sonicated for 10 min before transferring into vials equipped with inserts.

### 4.3. High-Performance Liquid Chromatography Tandem Mass Spectrometry (HPLC-MS/MS) Analysis

All analyses were performed on a Thermo Ultimate 3000 coupled with a high-resolution AB Sciex TripleTOF 5600 mass spectrometer. For the untargeted metabolomic analyses, all of the collected samples, together with quality-control samples and blank samples, were injected in both positive and negative (ESI+, ESI−) modes using the SWATH method. Samples were separated on a column (Kinetex C18, 2.6 μm × 150 × 3 mm column; Phenomenex). The column temperature was maintained at 30 °C. The mobile phase was composed as follows: A = 0.1% formic acid and B = 0.1% formic acid in MeOH. For both positive and negative modes, the linear elution gradient from 5% B (0–2 min) to 100% B (18–23 min) was applied, the initial gradient conditions were restored within 2 min (23–25 min), and the last 5 min of the HPLC method (25–30 min) were applied to maintain the beginning conditions. The flow rates were 220 μL/min^−1^, and the sample injection volume was 5 μL. Samples were held in an autosampler at 4 °C, and each sample was injected twice in two mass-window applications for the mass/charge (*m*/*z*) range (50–600 Da, 500–1200 Da). The ESI source conditions were set as follows: ion source gas 1 (GS1) 40 psi, ion source gas 2 (GS2) 40 psi, curtain gas (CUR) 35 psi, ion spray voltage 5500 V for ESI+, −4500 V for ESI−, and the source temperature was set for both modes to 400 °C.

The modified method was used for targeted analysis for the determination of small molecular metabolites. The separation was achieved on the column (Kinetex C18, 2.6 μm × 150 × 3 mm; Phenomenex) with the following eluent system: A = 0.1% HCOOH, 2.5 mM of Ammonium Formate, and B = 0.1% HCOOH in MeOH. A 2–12 min linear gradient (5–100% B) was used, with a constant flow rate of 0.22 mL/min^−1^. The injection volume was set to 5 μL. Samples were held in an autosampler at 4 °C, and each sample was injected twice for an *m*/*z* range of 50–500 Da. The ESI source conditions were set as follows: ion GS1, 40 psi; ion GS2, 40 psi; CUR, 35 psi; ion spray voltage, 5500 V; and source temperature, 400 °C.

Separation of fatty acids was carried out on a Kinetex C8, 2.6 μm, 100 × 2.1 mm column at 45 °C. The analytical column was equipped with a SecurityGuard Ultra C8 cartridge as a precolumn (internal diameter 2.1 mm). Mobile phase A consisted of 0.1% of acetic acid, and mobile phase B consisted of acetonitrile/methanol/water (80:15:5; *v*:*v*:*v*) with 0.1% acetic acid, with an injection volume of 5 μL. The following gradient was applied: 1 min—25% of B; 3 min—linear gradient 25–70% B; 8 min—linear gradient 70–90% B; 11 min—linear gradient from 90–100% B; 15 min—100% B; 17 min—25% B; 20 min—25% B; with a constant flow rate of 300 μL/min. Ionization was carried out in negative electrospray ionization (ESI−) mode with the following source settings: Gas1, 40 psi; Gas2, 40 psi; curtain gas, 35 psi; ion spray voltage, −4500 V; and temperature, 500 °C.

### 4.4. Data Processing

The LC-MS/MS data were processed using Sciex OS software (version 3.3, AB SCIEX, Toronto, ON, Canada). MarkerView software (version 1.3.1, AB SCIEX, Toronto, Canada) was used in the second step to process raw LC-HRMS data (peak detection, alignment, data filtering, and determination of the *m*/*z* ratio, RT and the ion peak area for each sample). Data mining was performed by the program algorithm—the peak intensity cut-off was set at 100 cps. Peak settings were achieved using retention time (RT) and mass to charge (*m*/*z*) with tolerances of 0.1 min and 0.005 Da, respectively. Monoisotopic peaks alone were considered to reduce mass abundancy and enhance the selection of a true molecular feature. Finally, mass signals differentially expressed by the control and case study samples (sclerosis multiplex) were identified by applying an additional filtering procedure with fold change (<1.5) and *t*-test (*p* > 0.05). This procedure is necessary for the elimination of the background and contaminants and preserved the true biological mass signals from the LC-HRMS data. The following steps were carried out using the MetaboAnalyst 5.0 web server. Acquired and filtered data from MetaboAnalyst 5.0 were, in the following step, verified with previously acquired data from targeted analyses (analyses of ~80 standards include amino acids, fatty acids and other small metabolites already set into the spectral library using Sciex OS software (version 3.3, AB SCIEX, Toronto, Canada).

### 4.5. Extraction of Clinical Data

Additionally, in 40 subjects after the first clinical symptoms of multiple sclerosis, we extracted clinical data from the database of the Center for Demyelinating Diseases and the Neurological Department of the Third Faculty of Medicine, Charles University and University Hospital Kralovske Vinohrady, Prague, Czech Republic. We took their clinical data in the form of Expanded Disability Status Scale (EDSS) [54] values from the time of CSF collection, and 1 and 2 years thereafter. All 40 subjects had EDSS values at the time of CSF collection. A total of 31 subjects had EDSS values 1 year after, and 22 subjects had EDSS values 2 years after the sample collection. All extracted data was anonymized and coded to protect personal patient information.

### 4.6. Statistical Analyses

We performed basic descriptive statistical analyses with follow-up of correlation analyses of clinical and metabolomic results using the free software “R” [55]. Student’s *t*-test, more precisely the two-tailed Welch t-test variant, was used to evaluate the statistical significance of the metabolomics comparison between patients after the first clinical symptoms of MS and the control group. In the next step, we applied the FDR (Benjamini–Hochberg false-discovery rate) for multiple comparisons to minimize false positive results. We evaluated the correlation coefficient (r), the significance of which was verified by using Student’s *t*-test. In all cases, our target *p*-value was estimated to be lower than 0.05, in an ideal case lower than 0.01.

## 5. Conclusions

Metabolomics, as a part of multiomics studies, presents a new source of biomarkers. In the case of MS, these new biomarkers could potentially help us to better understand the pathophysiology of the disease, and predict its course or even the efficacy of treatment. In this study, we have shown that there are statically significant differences in concentrations of different metabolites (mainly in the case of amino acids) in the CSF of MS patients, which may serve as new MS biomarkers. We can hypothesize that the level of arginine could reflect an active inflammatory process in MS, and levels of histidine may have some prognostic potential in the early stages of MS and be able to predict EDSS in the following few years. Glutamate may demonstrate the process of neurodegeneration in MS, choline may show the capacity of the organism to remyelination, and different fatty acids may reveal the processes of demyelination and remyelination. Spermidine, not yet described in MS, may point out anti-inflammatory capacities of the organism against autoimmune inflammation. These findings need to be verified in larger MS patient groups and compared to other neurological diseases with specificity for multiple sclerosis.

## 6. Study Limitations

One of the study limitations is a significant disproportion in the amount of male and female patients. We collected data from 57 females but only 16 males. This disproportion can be explained by several factors. First, MS is a disease that predominates among young women, 2–3 times more often than men [56]. Second, the willingness to look for medical help and undergo different examinations is more common in women, even in cases of smaller neurological impairment. In our study, women showed a greater willingness to give their consent and be part of the study than men.

A second study limitation is the smaller sample size because of the strict inclusion and exclusion criteria and following patient drop-out during the two-year follow-up period.

A third study limitation is that we did not have quantitative MRI data (number of brain and spinal cord lesions, volume of lesions, active lesions, etc.); therefore, we could not analyze the relationship between MR imagining and metabolomics results.

The last study limitation is that we focused only on MS patients with positive OCB findings in their CSF.

## Figures and Tables

**Figure 1 ijms-24-16271-f001:**
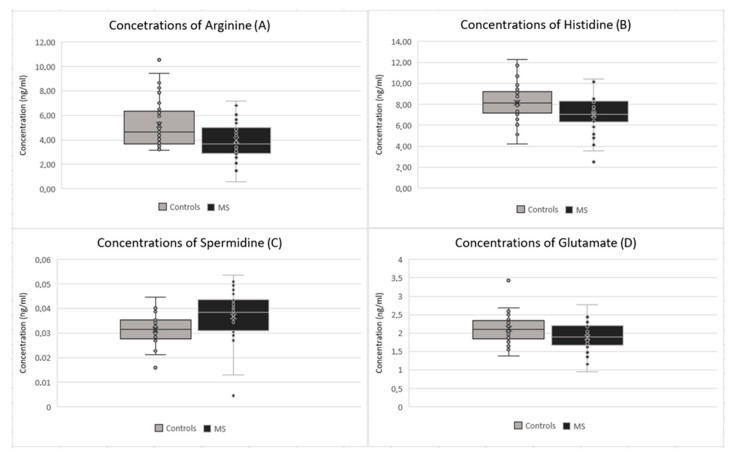
Comparison of concentrations between patients after the first clinical symptoms of MS and the control group in (**A**) arginine; (**B**) histidine; (**C**) spermidine; (**D**) glutamate.

**Figure 2 ijms-24-16271-f002:**
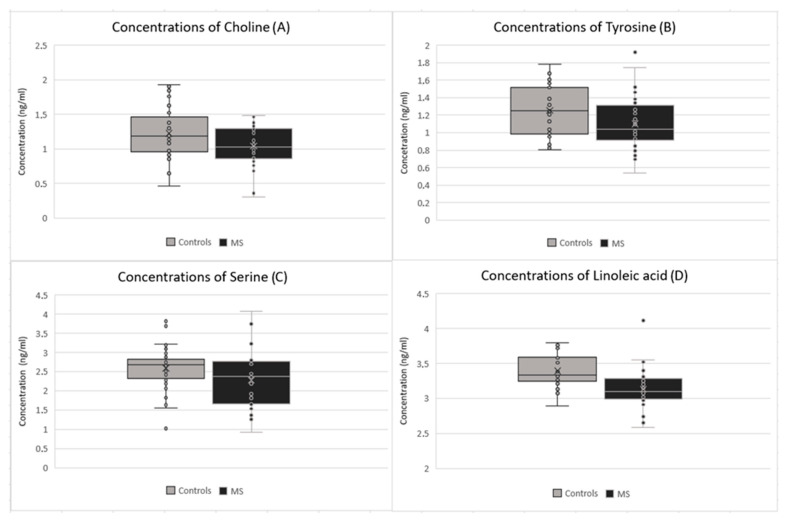
Comparison of concentrations between patients after the first clinical symptoms of MS and the control group in cases of (**A**) choline; (**B**) tyrosine; (**C**) serine; (**D**) linoleic acid.

**Figure 3 ijms-24-16271-f003:**
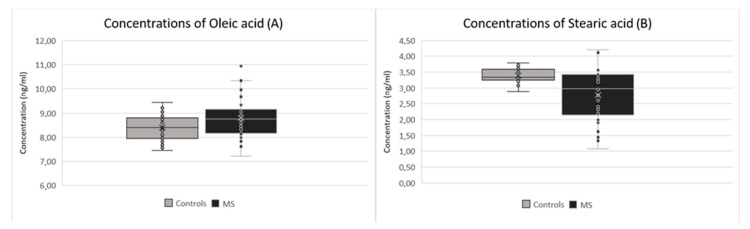
Comparison of concentrations between patients after the first clinical symptoms of MS and the control group in (**A**) oleic acid; (**B**) stearic acid.

**Figure 4 ijms-24-16271-f004:**
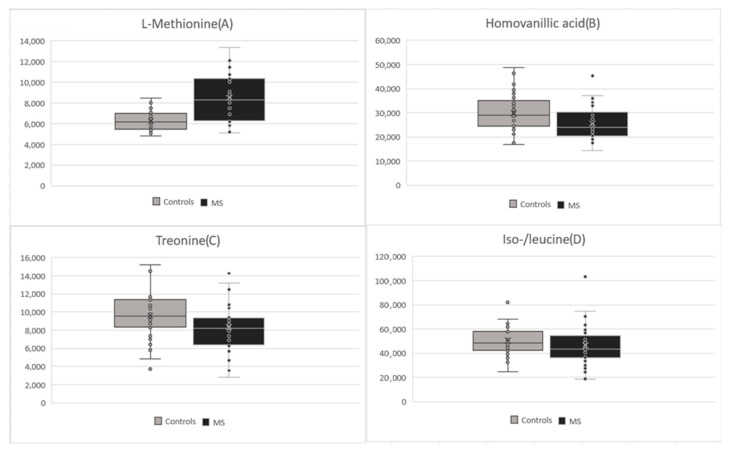
Comparison between patients after the first clinical symptoms of MS and the control group in (**A**) methionine; (**B**) homovanillic acid; (**C**) threonine; (**D**) iso-/leucine.

**Figure 5 ijms-24-16271-f005:**
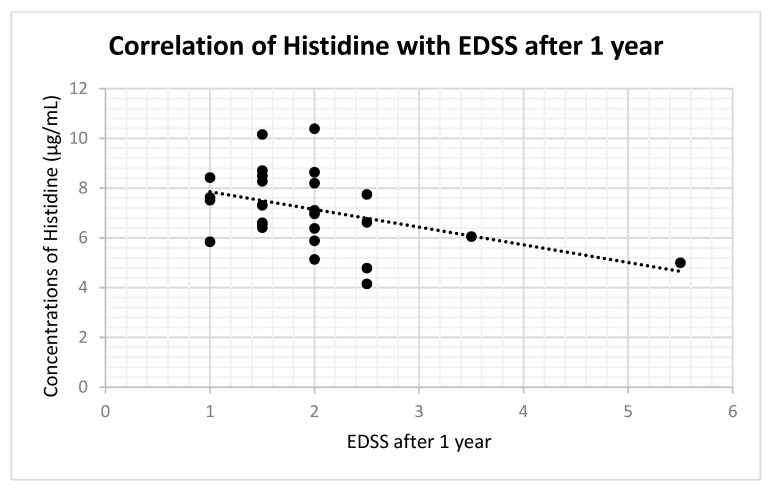
Correlation of histidine concentration with EDSS values after 1 year.

**Table 1 ijms-24-16271-t001:** Metabolomic results of chosen metabolites (concentrations in μg/mL).

Metabolites	Mean (±SD) MS (μg/mL)	Mean (±SD) C (μg/mL)	*p*-Value
Arginine	3.9708 (±1.4863)	5.2351 (±1.9583)	**0.0037**
Histidine	7.0202 (±1.6554)	8.1648 (±1.7159)	**0.0058**
Spermidine	0.0367 (±0.01111)	0.0313 (±0.0059)	**0.0124**
Glutamate	1.8883 (±0.3890)	2.1190 (±0.3840)	**0.0145**
Choline	1.0436 (±0.2841)	1.2276 (±0.3691)	**0.0233**
Tyrosine	1.1032 (±0.2936)	1.2527 (±0.2786)	**0.0313**
Serine	2.2714 (±0.7321)	2.5920 (±0.6051)	**0.0473**
Linoleic acid	3.1304 (±0.277)	3.3933 (±0.228)	**0.001**
Oleic acid	8.797 (±0.824)	8.393 (±0.523)	**0.015**
Stearic acid	2.7739 (±0.825)	3.1582 (±0.623)	**0.029**

Note: MS = MS patient group; C = control group; SD = standard deviation.

**Table 2 ijms-24-16271-t002:** Metabolomic results of other metabolites (comparison of peak areas).

Metabolites	Mean (±SD) MS	Mean (±SD) C	*p*-Value
Methionine	8504.86 (±2293.06)	6256.73 (±896.85)	**<0.001**
Homovanillic acid	25,611.18 (±6850.19)	29,774.24 (±8085.84)	**0.024**
Threonine	8363.92 (±2797.24)	9587.38 (±2447.98)	0.053
Uridine	54,333.68 (±37,916.23)	73,550.91 (±46,254.03)	0.063
Oxoglutaric acid	3671.48 (±585.98)	3466.02 (±400.76)	0.086
Serotonin	72,399.12 (±103,619.34)	36,146.67 (±70,506.17)	0.086
Biotin	18,633.25 (±10,638.00)	23,614.50 (±14,333.83)	0.106
Palmitoleic acid	16,785.92 (±2676.63)	15,807.12 (±2426.12)	0.111
Dopamine	7914.79 (±2566.33)	8832.67 (±2475.26)	0.130
Adipic acid	93,639.30 (±19,305.67	98,516.06 (±10,593.10)	0.185
Creatine	414,855.54 (±100,129.74)	444,733.33 (±98,329.90)	0.210
Tryptophane	75,840.66 (±12,870.37)	71,938.48 (±13,716.55)	0.223
Iso-/leucine	46,331.37 (±16,522.71)	50,641.21 (±12,977.94)	0.223
Adenine	208,212.43 (±44,745.61)	219,853.18 (±36,331.27)	0.231
Alanine	9162.12 (±2287.16)	9821.76 (±2585.61)	0.262
Uric acid	18,416.67 (±11,654.81)	16,896.35 (±8212.53)	0.523
Cystine	13,257.83 (±969.36)	13,426.36 (±1277.90)	0.539
Aspartate	5438.97 (±918.10)	5558.92 (±726.05)	0.541
Palmitic acid	53,262.32 (±20,906.23)	54,647.30 (±8844.53)	0.712
Myristic acid	4917.08 (±2478.17)	5009.62 (±1479.14)	0.847

Note: MS = MS patient group; C = control group; SD = standard deviation.

**Table 3 ijms-24-16271-t003:** Correlation of metabolites and EDSS values.

Metabolites	r EDSS 0	p-v EDSS 0	r EDSS 1	p-v EDSS 1	r EDSS 2	p-v EDSS 2
Histidine	0.043	0.744	−0.415	**0.031**	−0.437	**0.048**
Arginine	−0.003	0.577	−0.104	0.604	−0.023	0.920
Spermidine	−0.163	0.329	−0.324	0.087	0.005	0.982
Glutamate	−0.212	0.202	−0.066	0.736	0.099	0.670
Choline	−0.161	0.168	−0.288	0.130	0.080	0.729
Tyrosine	−0.177	0.288	−0.125	0.519	0.034	0.883
Serine	−0.150	0.375	−0.066	0.733	0.042	0.857
Oleic a.	−0.056	0.740	−0.202	0.292	−0.036	0.877
Stearic a.	−0.202	0.224	−0.051	0.792	0.369	0.100
Linoleic a.	−0.016	0.925	0.057	0.771	0.091	0.695

Note: r = correlation coefficient; *p*-v = *p*-value; EDSS = Expanded Disability Status Scale; a. = acid.

## Data Availability

The data that support the findings of this study are available on request from the corresponding author, I.Š. The data are not publicly available due to their containing information that could compromise the privacy of research participants.

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
