# Peer review of "Metabolomics of Cerebrospinal Fluid Amino and Fatty Acids in Early Stages of Multiple Sclerosis"

_ijms, 2023, doi:10.3390/ijms242216271_

Round 1
Reviewer 1 Report
Comments and Suggestions for Authors
The authors cover a hot topic in MS and autoimmune conditions in general. Their utilized methodology is adeqaute, the results are presented clearly, the drawn conclusions are fine but not too far-reaching. All in all, I recommend the article for publication.
Author Response
Thank you for your review.
Reviewer 2 Report
Comments and Suggestions for Authors
The manuscript is purely descriptive with no suggestion of what the metabolomics mean. Which CSF proteins are important? The data do not support the author’s conclusion until it is shown that the present values are found ONLY or MOSTLY in MS patients. The work provides an interesting start to the research, but clearly not the end result.
Follow up study at years 1 and 2 is an excellent idea. Population size is appropriate as are the methods. The interpretation needs to be addressed.
Was the CSF obtained without anesthesia?
Tables should have means included. Standard deviations for some of the metabolites are huge. What are the units in table 1 and table 2?
Discussion is lengthy and should be reduced by 50%. Commentary on factors that were not significantly different should be reduced to a few sentences each.
Reduce the reference list by 30%.
Author Response
The manuscript is purely descriptive with no suggestion of what the metabolomics mean. Which CSF proteins are important? The data do not support the author’s conclusion until it is shown that the present values are found ONLY or MOSTLY in MS patients. The work provides an interesting start to the research, but clearly not the end result.
Answer: Thank you for your review. Our study primarily aimed to identify variances in metabolites within the cerebrospinal fluid of individuals in the initial phases of multiple sclerosis as compared to those who are healthy through untargeted approach. Therefore, we did not have prechosen proteins or metabolites, but found them through the untargeted metabolomic analyses and after that we aimed to elucidate the underlying reasons for these alterations from a pathophysiological perspective. We did not focus on other diseases, that would be the aim of following studies.
Follow up study at years 1 and 2 is an excellent idea. Population size is appropriate as are the methods. The interpretation needs to be addressed.
Was the CSF obtained without anesthesia?
Answer: The CSF was obtained without any form of anesthesia.
Tables should have means included. Standard deviations for some of the metabolites are huge. What are the units in table 1 and table 2?
Answer: Tables were corrected. Units in the first table are in ng/mL; the second table, due to limited access to the deuterated standards required for concentration determination, are only the result of peak area comparisons. This is the main reason why the standard deviation values are so huge.
Discussion is lengthy and should be reduced by 50%. Commentary on factors that were not significantly different should be reduced to a few sentences each. Answer: Discussion was reworked and shortened.
Reduce the reference list by 30%.
Answer: We reduced the reference list, but we were not able to reduce it further. We consider rest of the references essential for manuscript.
Reviewer 3 Report
Comments and Suggestions for Authors
The authors have dealt with a very intriguing subject of the Neurological community: MS biomarkers.
The execution of the study is very well described and besides the difficulties due to patients' drop-out, they managed to have data for a 2-year duration.
Some minor comments and corrections are needed:
1. Line 39 "economic impact": Reference is needed.
2. Line 59 "sick patients": Please omit the word sick. Not appropriate.
3. Lines 111-115 "Patients from ... neurological diseases": What did the MRI show in those patients and under what reasoning did you perform LP for CSF analysis? Did they have non-specific lesions?
4. Line 212 "R software": Please add reference "R Core Team (2021). R: A language and environment for statistical computing. R Foundation for Statistical Computing, Vienna, Austria. URL https://www.R-project.org/."
5. Line 328 "CSP": Do you mean CSF?
6. References: Please correct the references. They are shown with double the enumeration.
7. Make a separate limitation paragraph.
8. You mentioned that all patients had positive OCBs. Do you have any data that support these findings also in patients with negative OCBs? If not write it in the limitation section.
9. Do you have data regarding the existence of lesions in the spinal cord compared to patients without SC lesions? Please elaborate on that.
10. You mention the disease activity in the discussion section regarding L-arginine. Do you have data from those patients regarding the active (enhancing) MRI lesions? If not, add a comment in the limitation paragraph.
Thank you.
Author Response
Thank you for your review.
1. Answer: It was corrected.
2. Answer:: It was corrected
3. Answer: Brain MRI findings were normal or had few non-specific lesions. The indication for lumbar puncture was to rule out subarachnoid bleeding or neuroinfectious diseases (based on common headache and non-specific symptoms).
4. Answer: Reference was added.
5. Answer: It was corrected
6. Answer: It was corrected
7. Answer: Paragraph was added.
8. Answer: It was added into limitations.
9. Answer: It was added into limitations.
10. Answer: It was added into limitations.
Reviewer 4 Report
Comments and Suggestions for Authors
The manuscript “Metabolomics of Cerebrospinal Fluid Amino and Fatty Acids in Early Stages of Multiple Sclerosis” (ijms-2569343) by Židó et al contains interesting perspectives of cerebrospinal fluid (CSF) metabolites concentration as biomarker. Manuscript is well written and authors have shown in depth knowledge of subject. I have a few major concerns listed below-
1- First concern is related to presentation of results in figure 1-4, authors are suggested to present the results in consistent manner. As there are scattered dots on control group bar while Multiple Sclerosis (MS) group bar doesn’t have scattered dots.
2- What is the rationale to compare the standard deviations of metabolites of MS and Control groups in table 1 and 2?
3- Authors are suggested to update the table 1 and 2 with Mean ± SD and significance p-value of mean values of MS and Control groups.
Author Response
Thank you for your review.
1- Answer: It was corrected
2- Answer: Tables were reworked.
3- Answer: Tables were reworked.
Round 2
Reviewer 2 Report
Comments and Suggestions for Authors
Authors have addressed the comments.
Author Response
Thank you for your review!
Reviewer 4 Report
Comments and Suggestions for Authors
The authors have successfully addressed a majority of the issues previously raised. However, there remain several critical points that must be rectified before the manuscript is fit for publication-
Why authors are changing the concentration unit from microgram to nanogram? In your previously published manuscript (refer to the reference number 26, DOI: 10.3389/fneur.2022.874121), metabolites concentrations was displayed in microgram unit.
In your previous study (as mentioned above), Arginine and Histidine concentration in figure 1 was reduced in the MS patients. In this manuscript authors are showing, Arginine and Histidine concentration in figure 1 is increasing in the MS patients. While there is no change in the number of patients. Authors needs to justify.
Author Response
Why authors are changing the concentration unit from microgram to nanogram? In your previously published manuscript (refer to the reference number 26, DOI: 10.3389/fneur.2022.874121), metabolites concentrations was displayed in microgram unit.
Answer: Thank you for your point, this was an overlooked mistake from our part. We have double-check and correct units are microgram - it was corrected in newest version of manuscript. Thank you again for your reminder!
In your previous study (as mentioned above), Arginine and Histidine concentration in figure 1 was reduced in the MS patients. In this manuscript authors are showing, Arginine and Histidine concentration in figure 1 is increasing in the MS patients. While there is no change in the number of patients. Authors needs to justify.
Answer: I am sorry if I dont understand this part correctly but in both (previous publication and current manuscript) concentrations of both histidine and arginine were decreased in MS patients compared to controls (as shown in figures, tables and text). In our previous publication we had 19 MS patients and 19 controls and in our current manuscript, there are 40 MS patients and 33 controls, so the number of patients nearly doubled).
Thank you for your review!
Round 3
Reviewer 4 Report
Comments and Suggestions for Authors
Thank you for your response. Second concern arise due to swapping the position of groups in your current manuscript vs previously published. Well there is no such rule which recommends to keep control group first followed by the unhealthy group. In general, everyone follows this pattern.
Author Response
Thank you for your comment. This was changed in the latest version of our manuscript.